# Effects of *Lactiplantibacillus plantarum* GUANKE on Diphenoxylate-Induced Slow Transit Constipation and Gut Microbiota in Mice

**DOI:** 10.3390/nu15173741

**Published:** 2023-08-26

**Authors:** Yuanming Huang, Yanan Guo, Xianping Li, Yuchun Xiao, Zhihuan Wang, Liqiong Song, Zhihong Ren

**Affiliations:** National Key Laboratory of Intelligent Tracking and Forecasting for Infectious Diseases Chinese Center for Disease Control and Prevention, National Institute for Communicable Disease Control and Prevention, Chinese Center for Disease Control and Prevention, Beijing 102206, China; huangyuanming@icdc.cn (Y.H.); 13561062802@163.com (Y.G.); qdlixianping@126.com (X.L.); xiaoyuchun@icdc.cn (Y.X.); lnsffxwzh@126.com (Z.W.)

**Keywords:** slow transit constipation, *Lactiplantibacillus plantarum*, probiotic, intestinal microbiota, *Prevotella*

## Abstract

Slow transit constipation (STC) is a prevalent gastrointestinal condition with slow transit, and some probiotics can effectively relieve constipation, but the exact mechanisms have not been fully understood. In this study, we evaluate the impact of *Lactiplantibacillus plantarum* GUANKE (GUANKE) on diphenoxylate-induced slow transit constipation and speculate on the underlying mechanisms in a mouse model. Administration of *L. plantarum* GUANKE alleviated constipation indexes, including defecation time, fecal output and water content, and gastrointestinal transit ratio. In addition, GUANKE restored the protein expression of constipation-related intestinal factors (aquaporins (AQPs) and interstitial Cajal cells (ICCs)) in colon tissues measured using immunofluorescence staining; regulated the neurotransmitters and hormones, such as increased levels of 5-hydroxytryptamine, substance P, and motilin; and decreased levels of vasoactive intestinal peptide and nitric oxide in serum, as measured by an ELISA. 16S rRNA and correlation analysis of feces indicated that GUANKE administration effectively reduced constipation-induced *Prevotella* enrichment and suggested a potential contribution of *Prevotella* to diphenoxylate-induced STC in mice. GUANKE had no effect on short-chain fatty acids (SCFAs) in cecum content. This study revealed that GUANKE may alleviate constipation in mice through regulating intestinal neurotransmitter and hormone release and altering specific bacterial taxa, rather than by affecting SCFAs and the diversity of microbiota in the gut. Further research is needed to confirm if the findings observed in this study will be consistent in other animal studies or clinical trials.

## 1. Introduction

Slow transit constipation (STC) arises from a complex interplay of factors that converge to disrupt the normal colonic motility and stool transit process. The hallmark of STC is the reduced rate of colonic motility, resulting in sluggish stool movement and a range of distressing symptoms. This condition is marked by the presence of dry, difficult-to-pass stools, along with sensations of incomplete or infrequent defecation, often accompanied by abdominal distension [1,2,3,4]. Understandably, the far-reaching impact of STC on a patient’s quality of life cannot be underestimated. The prevalence rate of STC in China is about 6% but has been rapidly increasing; STC incidence is correlated with age, gender, and dietary habits, among others [5,6]. Although the trigger factors for STC are not clear, it has been reported that intestinal neuropathy, impaired neurotransmission, and abnormal hormonal effects play important roles in the development of STC [7,8,9,10,11]. Specifically, intricate neuropathic alterations within the intestinal environment contribute to the compromised transit that characterizes STC. Impaired neurotransmission pathways, critical for coordinated bowel movement, prevent the necessary rhythmic contractions. 5-hydroxytryptamine (5-HT), a key enteric and central nervous system neurotransmitter, regulates motility through secretion and the initiation of peristalsis [12]. In addition, hormonal imbalances disrupt colonic coordination by interfering with enteric nervous system communication. The onset of constipation also correlates with decreased water channel proteins and interstitial cells of Cajal, known as the pacemakers of gastrointestinal peristalsis. Common treatment includes laxatives, sacral nerve stimulation, and biofeedback, aiming to restore normal intestinal motility and relieve the symptoms of constipation, but they are questioned for long-term use [13]. Thus, it is important to develop alternative treatments for STC.

Recent studies have revealed that the intestinal microbiota plays a crucial role as an endocrine organ to be involved in regulating host physiological activities including gastrointestinal motility [12,14,15]. Probiotics are the live microorganisms that, when consumed in appropriate amounts, confer health benefits to the host. Their potential to modulate intestinal function and improve overall health has prompted interest in their application for managing constipation and other gastrointestinal disorders [16]. Studies have documented the utilization of various probiotics, prebiotics, or combined formulations known as synbiotics in both animal experiments and clinical trials to alleviate constipation [17,18,19,20,21]. *Lactiplantibacillus plantarum* is commonly used alone or in combination with other probiotics to relieve constipation [22,23]. Several potential mechanisms have been proposed to elucidate the beneficial effects of *L. plantarum* on constipation, including the modulation of gut microbiota composition, regulation of intestinal motility and secretion, and an improvement in gut barrier function. *L. plantarum* KFY02 was reported to alleviate constipation induced by a low-fiber diet in mice by regulating the abundance, diversity, and structure of the intestinal microbiota toward homeostasis [23]. In a randomized, double-blind, placebo-controlled study, *L. plantarum* Lp3a conferred significant improvements in functional constipation by affecting the potential biological mechanisms including fatty acid metabolism, methane, and bile acid biosynthesis, although it had no significant effect on the gut microbiota [18]. Other probiotics, such as *Lacticaseibacillus casei*, *Lactobacillus acidophilus*, *Lacticaseibacillus Rhamnosus*, *Lactobacillus paraccasei*, *Bifidobacterium animalis subsp. lactis*, *Bifidobacterium animalis*, and *Bacillus subtilis,* have also been investigated for their potential function in relieving constipation [20,22,24,25]. While these probiotics have protective effects on constipation, the underlying mechanisms need to be further explored.

*Lactiplantibacillus. plantarum* GUANKE (GUANKE) is a novel strain which was isolated from the feces of a healthy volunteer and has probiotic potential including tolerating bile and acidic environments in the intestinal tract [26]. In this study, we systemically evaluated the effects of oral administration of *L. plantarum* GUANKE on constipation symptoms, water channel protein expression, and the SCF/c-kit pathway in colon tissues via immunohistochemistry and the quantification of neurotransmitters and hormones in serum, short-chain fatty acid abundance in cecum content, and fecal gut microbiota using a diphenoxylate-induced constipation mouse model, which may provide evidence for the utilization of probiotics in preventing constipation.

## 2. Materials and Methods

### 2.1. Animals and Strain

To assess the alleviating effect of *L. plantarum* GUANKE on constipation, 18 male ICR mice (6- to 7-weeks-old, 24–26 g, Beijing Vital River Laboratory Animal Technology, Beijing, China) were acclimatized in the Animal Center of China CDC for one week prior to use. All experimental protocols were approved by the Ethics Review Committee of the National Institute for Communicable Disease Control and Prevention at the Chinese Center for Disease Control and Prevention (2020-025).

*L. plantarum* GUANKE strain was isolated from the feces of a healthy volunteer and was cultured in MRS medium at 37 °C for 18 h. Cell pellets were centrifuged for 10 min at 1000× *g*, and washed with sterile PBS, then reached 1 × 10^10^ CFU/mL for intragastrical administration.

### 2.2. Establishment of Constipation Mouse Model and L. plantarum GUANK Intervention

The STC mouse model was induced by diphenoxylate (H22022037, Changhong Pharmaceutical, Changchun, China). The ink discharge was utilized as a marker to indicate the passage of the first black stool. The ink was prepared freshly every time prior to use by adding 10 g of activated carbon powder (C7261, Beijing Prince Technology, Beijing, China) to the boiling Arabic solution. After a week of acclamation, the mice were divided into three groups: normal, constipation, and GUANKE treatment, with six animals in each group. The normal and the constipation groups were administered 0.3 mL of PBS daily in the morning, and GUANKE treatment groups were given 0.3 mL of alive *L. plantarum* GUANKE (3 × 10^9^ CFU each mouse) intragastrically. Following 21 days of intervention, the mice were kept on a 16 h fast. On day 22, all mice were given diphenoxylate (10 mg/kg) intragastrically except for the normal group (Figure 1a). At 30 min after treatment, all mice were gavaged with 0.25 mL of ink, then placed individually in clean, sterilized cages and given water and food. The feces from each mouse were collected within 5 h after oral intake of the ink. Indicators of defecation were recorded for each mouse, which included the time of starting to expel black stool for the first time, total weights, and numbers of stool. The fecal samples were partitioned into three equal portions to assess fecal water content, 16S rRNA sequencing, and short-chain fatty acids (SCFAs). The collected feces were dried in an oven at 100 °C for 2 h to remove moisture, and the dry weight of feces was measured to calculate the fecal water content using the following formula [27].
fecal water content = (Wet Weight − Dry Weight)/(Wet Weight) × 100%

### 2.3. Measurement of Intestinal Transit Ratio in Mice

The assessment of intestinal transit ratio was conducted as described previously [17]. Briefly, after the experimental period, all mice subjected to the above treatments were kept fasted for 16 h. Then, 0.3 mL of diphenoxylate was orally administered to all mice except the normal group. After 25 min, all mice underwent cervical dislocation following carbon dioxide euthanasia. The entire bowel was removed, and its length was measured. The gastrointestinal transit percentage was determined by measuring the distance from the pylorus to the front end of the ink (D_ink_) and dividing it by the total length of the intestinal tract (D_total_). The intestinal transit ratio was calculated using the following equation:intestinal transit ratio = D_ink_/D_total_ × 100%

### 2.4. Immunohistochemistry Examination of AQP4, AQP8, C-Kit, and SCF

Immunohistochemistry experiments were conducted to assess the expression of AQP4, AQP8, and the SCF/c-kit pathway in the colon, following the previously described methods [17]. Briefly, the section of the colons near the cecum was collected and preserved in 4% formaldehyde for fixation, degreased in xylene, then washed in alcohol, and finally incubated with 5% BSA (Wuhan Boster, Wuhan, China) for 30 min. After incubating the samples with primary antibodies overnight at 4 °C, they were rewarmed for 1 h at RT and washed three times with PBS. Subsequently, the secondary antibody (goat anti-rabbit IgG, Beyotime, Haimen, China) was added and incubated for 30 min at 37 °C followed by three washes with PBS. Next, DAB color development was performed, and hematoxylin was used for counterstaining, differentiation, and anti-blue staining. After the colon tissues underwent final dehydration, washing, and mounting, their images were examined and recorded under a microscope and the mean optical density (MOD) values were calculated. The primary antibodies included anti-AQP4 antibody (Abcam, Boston, MA, USA), anti-AQP8 antibody (Absin, Shanghai, China), anti-c-Kit antibody (CST, Beverly, MA, USA), and anti-SCF antibody (Absin, China), and the tissue sections were captured using a Computer Image Processing System equipped with a CMOS camera (OLYMPUS, Tokyo, Japan). The images were then analyzed using Image-Pro Plus 6.0 software (Media Cybernetic, Rockville, MD, USA). The positive immunostaining intensity and the corresponding tissue area were assessed through a combination of human–computer interaction. MOD was derived by calculating the ratio of integrated optical density to the area of tissue. These outcomes were evaluated by a trained individual who was unaware of the treatment conditions associated with the samples. The statistical analysis of the average optical density was performed using GraphPad Prism 5.

### 2.5. Measurement of SCFAs

The SCFAs were measured as described previously [17]. Briefly, the cecum contents of the mice were dissolved with 0.5% phosphoric acid and subjected to centrifugation at 14,000× *g* for 10 min to gain the supernatant. The resulting supernatant was then mixed with an equal volume of ethyl acetate and centrifuged again. The upper organic phase was collected for GC-MS analysis. The upper sample volume in GC-MS analysis was 1 μL with a separation ratio of 10:1. The initial temperature was set at 90 °C, sequentially ramped up to 120 °C at a rate of 10 °C/min, then raised to 150 °C at a rate of 5 °C/min and finally to 250 °C at a rapid rate of 25 °C/min, where it was maintained for 2 min.

### 2.6. Quantification Measurement of Serum Neurotransmitters and Hormones

The expression of substance P (SP), 5-hydroxytryptamine (5-HT), nitric oxide (NO), vasoactive intestinal peptide (VIP), gastrin (GAS), motilin (MTL), and endothelin (ET) in serum was measured using ELISA kits (Beijing Yisheng Zhaobo Biotechnology, Beijing, China) following the instructions of the manufacturer.

### 2.7. 16S rRNA Sequencing and Gut Microbiota Analysis

The total bacterial genomic DNA of the fecal samples was extracted using MagPure Stool DNA KF kit B (Magen, Guangzhou, China); a quality assessment was performed using agarose gel electrophoresis and quantified by the Qubit™ dsDNA BR Assay Kit (Invitrogen, Waltham, MA, USA). Universal forward (341-F: 5′-ACTCCTACGGGAGGCAGCAG-3′) and reverse (806-R: 5′-GGACTACHVGGGTWTCTAAT-3′) primers were used for the amplification of the V3-V4 hypervariable regions of the bacterial 16S rRNA. The libraries were constructed via the Agilent 2100 Bioanalyzer (Agilent, Santa Clara, CA, USA) and the sequencing was performed on the Illumina Novaseq platform. Quality control was performed by FastQC and filtered for bases with Phred scores <20. Clean data were merged and filtered of low-quality reads; then, chimera removal was performed [28], and operable taxonomic unit (out) construction was performed using both USEARCH (V10.0.240) [29] and VSEARCH (V2.8.1) [30]. Alpha and beta diversity indices were computed by USEARCH (V10.0.240), and principal co-ordinates analysis (PCoA) was calculated by R software (V3.6.1) (amplicon package, https://github.com/microbiota, accessed on 28 May 2023). The identification of differential bacteria between groups was determined by LEfSe (http://huttenhower.sph.harvard.edu/galaxy/, accessed on 28 May 2023), Wilcoxon rank-sum tests, and/or nonparametric factorial Kruskal–Wallis.

### 2.8. Statistical Analysis

IBM SPSS Version 21.0 was used for data analysis, and a *p* value < 0.05 was considered statistically significant. Comparisons between groups were determined by one-way analysis of variance (ANOVA) and Dunnett’s multiple comparisons test. The rank-sum test of the nonparametric test was used for a relative abundance analysis of the gut microbiota. A Pearson correlation analysis of fecal parameters and intestinal factors was performed. Data are presented as means ± standard deviation (SD).

## 3. Results

### 3.1. Effects of L. plantarum GUANKE on Fecal Parameters and Gastrointestinal Transit

To explore the effect of *L. plantarum* GUANKE on diphenoxylate-induced STC mice, we evaluated the fecal parameters and gastrointestinal transit ratio. Compared to the normal mice, the mice in the constipation group exhibited longer defecation times expelling the first black stool, which indicated a weaker intestinal motility. However, administration with *L. plantarum* GUANKE significantly reduced the time compared with the constipation group. Additionally, the feces weight, feces number, and fecal water contents were decreased significantly in the constipation group compared to the normal mice, and treatment with *L. plantarum* GUANKE increased the feces number and water content but had no effect on the feces weight (Figure 1c–e). Moreover, the gastrointestinal transit ratio in the constipation group decreased significantly compared to that in normal group, whereas *L. plantarum* GUANKE treatment led to a significant increase in the gastrointestinal transit ratio (Figure 1f). These results demonstrated that *L. plantarum* GUANKE may have potential as a probiotic to improve STC.

### 3.2. Effects of L. plantarum GUANKE on Protein Expression Levels of Constipation-Related Intestinal Factors

To explore the probiotic potential of *L. plantarum* GUANKE on the damage of the chemical barrier in STC mice, we first focused on the alteration of aquaporins (AQPs) in colonic epithelial cells. AQP4 and AQP8 are crucial regulators of colonic water metabolism, and their upregulation has been associated with increased water absorption, leading to reduced stool moisture and dry stool. The immunofluorescence staining results indicated that in the colon of constipated mice, the expression of AQP4 and AQP8 was significantly higher compared with that in the normal group, while the administration of *L. plantarum* GUANKE significantly reduced these proteins’ content in colonic epithelial cells (Figure 2a–c). Furthermore, we examined the c-kit receptor on the surface of interstitial Cajal cells (ICC) and the stem cell factor (SCF) associated with this pathway, considering that low expression of both affects intestinal motility. The immunohistochemical results showed that the levels of c-kit and SCF proteins were significantly lower in the constipation group than in the normal group, and the administration of *L. plantarum* GUANKE significantly increased the levels of these two proteins. These results suggested that treatment with *L. plantarum* GUANKE restored the protein levels of constipation-associated intestinal factors.

### 3.3. Effects of L. plantarum GUANKE on Serum Neurotransmitter and Hormone Levels

To examine the potential of *L. plantarum* GUANKE in modulating intestinal function by affecting intestinal neurotransmitter content, we assessed the levels of excitatory neurotransmitters, including SP and 5-HT, as well as inhibitory neurotransmitters, including VIP and NO. Our results demonstrated that the constipation group had significantly decreased levels of SP and 5-HT and significantly increased levels of VIP and NO compared the normal group. Notably, treatment with *L. plantarum* GUANKE resulted in a significantly higher level of SP and 5-HT, whereas the levels of VIP and NO decreased (Figure 3a–d). In addition, we investigated changes in intestinal hormones, including MTL, GAS, and ET. The results revealed that MTL and GAS in the constipation group were significantly lower, while ET was significantly higher compared with the normal group. The MTL levels, however, were significantly restored after treatment with *L. plantarum* GUANKE (Figure 3e–g). These findings indicated that *L. plantarum* GUANKE may serve as a promising therapeutic agent for the treatment of slow transit constipation by modulating intestinal neurotransmitters and hormones.

### 3.4. Effects of L. plantarum GUANKE on the Diversity of Intestinal Microbiota

To investigate whether *L. plantarum* GUANKE’s probiotic effect on alleviating constipation in STC mice was mediated by modulating the intestinal microbiota, we conducted 16S rRNA sequencing to analyze the composition and diversity of the intestinal microbiota. Our results indicated that the constipation group had a decreased abundance in Firmicutes and an increased abundance in Bacteroidetes compared to the normal group. Treatment with *L. plantarum* GUANKE restored this trend, although no significant statistical difference was found in the composition and diversity of the gut microbiota among the three groups (Figure 4a). Interestingly, LEfSe analysis identified that the relative abundance of the genus *Prevotella* was significantly higher in the intestines of STC mice than that in the normal group (*p* < 0.001). Treatment with *L. plantarum* GUANKE decreased the relative abundance of *Prevotella* (*p* < 0.01) (Figure 4b). We then explored whether *Prevotella* was associated with fecal parameters, gastrointestinal transit, intestinal neurotransmitters, and hormones by performing Pearson correlation analysis. The results suggested that the time of the first black stool defecation was positively associated with *Prevotella*, while the number, weight, fecal moisture content, and gastrointestinal ratio were negatively associated with *Prevotella*. The level of excitatory neurotransmitter 5-HT was negatively associated with *Prevotella*, while VIP and NO were positively correlated with *Prevotella*. The ET was positively correlated with *Prevotella*. However, no association was observed between MTL and *Prevotella*. These data suggested that the relief of slow transit constipation by *L. plantarum* GUANKE may be achieved by decreasing the relative abundance of *Prevotella* rather than affecting the diversity and richness of gut microbiota.

### 3.5. Effects of L. plantarum GUANKE on SCFA Profile in Mice

SCFAs are important metabolites resulting from the fermentation of dietary fibers by gut microbiota, which contribute to relieving constipation [31,32]. We evaluated fecal levels of SCFAs, including acetic acid, propionic acid, butyric acid, isobutyric acid, valeric acid, isovaleric acid, and hexanoic acid, to determine the involvement of SCFAs in the observed effects of *L. plantarum* GUANKE. We only found a higher level of acetic acid in the constipated mice than the normal group, while the administration of *L. plantarum* GUANKE had no effect on SCFAs (Appendix A). These results indicated that SCFAs may not be engaged in the relief of slow transit constipation by *L. plantarum* GUANKE, which instead may be due to the lack of a significant alteration in the diversity of gut microbiota.

## 4. Discussion

Constipation is a prevalent gastrointestinal disorder typically characterized by delayed intestinal transit time and difficulty in defecation [33]. However, commonly used medications such as laxatives have significant side effects [34], highlighting the need for alternative interventions. Some probiotics have emerged as a promising intervention for alleviating constipation, with many positive results demonstrated in human and animal studies [22,35,36,37]. Despite this, the exact underlying mechanisms are still poorly understood [38]. This study revealed that *L. plantarum* GUANKE treatment effectively improved defecation volume, fecal moisture content, and intestinal motility while reducing the secretion of AQP4 and AQP8 and restoring neurotransmitter and hormone secretion in the colon. These results suggest that *L. plantarum* GUANKE may exert a regulatory effect on slow transit constipation (Figure 5).

STC can be triggered by multiple factors, including the damage of the intestinal mucosal barrier, which affects intestinal peristalsis, the intestinal nervous system, and the balance of intestinal microbiota. In studies on probiotic intervention in STC, the enteric nervous system and gastrointestinal hormones are the most frequently evaluated indicators, such as SP, VIP, NO, somatostatin (SS), 5-HT, MTL, GAS, and ET. Of these markers, 5-HT is an important regulator which can modulate intestinal motility, bone development, and immune response [39]. Mosapride, as a novel 5-hydroxytryptamine 4 receptor (5-HT4R) agonist, is used to relieve constipation because of its ability to enhance gastrointestinal motility [40]. Our study showed that *L. plantarum* GUANKE reversed the reduction in 5-HT in constipated mice while at the same time significantly inhibiting the alteration in SP, VIP, NO, and MTL and restoring these indicators in STC mice to those of normal mice. Similarity, *L. plantarum* CQPC02, *L. plantarum* YS-3, *L. plantarum* KSFY06, and *L. sakei* Furu2019 have also been reported to have a positive ameliorative effect on the enteric nervous system and gastrointestinal hormones in constipation [17,41,42,43]; however, in comparison, GUANKE seems to be more efficient than those probiotics in restoring the levels of SP, 5-HT, and MTL.

Many studies have shown that alterations in the gut microbiota are closely associated with the pathogenesis of functional gastrointestinal disorders, including constipation [44,45]. Several clinical studies have revealed that the gut microbiota composition differs between constipated groups and healthy controls [46,47,48]. Compared to normal groups, the alpha diversity such as the Shannon index or OTU numbers is usually decreased in constipation groups [49]. In one study, the relative abundance of *Verrucomicrobia* in the feces of patients or mice with slow transit constipation was increased and *L. rhamnosus* CCFM1068 treatment reversed the alterations of *Verrucomicrobia* in the STC group toward the levels of the control group [14,38]. An increased abundance of *Akkermansia* in the constipation group was also observed in another animal study, and *L. plantarum* CCFM405 or CCFM1068 could reduce the relative abundance of *Akkermansia* [38,50]. Unexpectedly, in our study, decreased diversity in the gut microbiota was not observed, but increased levels of *Prevotella* were observed upon LEfSe analysis in constipated mice, and the treatment of *L. plantarum* GUANKE reversed this change caused by constipation. In accordance with our results, the increased abundance of *Prevotella* was also observed in a drug-induced STC rat model [51]. *Prevotella*, a Gram-negative anaerobic bacterium, is highly prevalent in the human gut microbiome and has been implicated in influencing human health through intricate ecological interactions and host crosstalk [52]. The predominant *Prevotella* species in the human gut consist of *Prevotella copri*, *Prevotella stercorea*, and closely related lineages [53]. Transplanting fecal samples containing an abundance of *P. copri* from RA patients into mice that were prone to rheumatoid arthritis induced a proinflammatory Th17 cell response and a phenotype resembling rheumatoid arthritis [54]. Several studies have mentioned the association between *Prevotella* spp. and dietary patterns as well as cardiometabolic health, but the conclusions are still controversial [55,56,57,58,59,60]. A study of constipation relief in patients with peritoneal dialysis by using Tiaopi Xiezhuo decoction (TXD) reported that *Prevotella* was enriched in TXD treatment, suggesting a positive correlation with an improvement in constipation, but its positive correlation with the serum levels of phosphate also suggests a potential health risk [61]. However, in our study, the correlation analysis demonstrated a strong association between *Prevotella* and defecation volume, fecal water content, intestinal transport rate, 5-HT, and NO, respectively, which indicates that a reduced abundance of *Prevotella* may be involved in relieving STC in mice treated by *L. plantarum* GUANKE. Additional investigations are required to fully comprehend the specific role of *Prevotella* spp. in alleviating STC.

## 5. Conclusions

In summary, *L. plantarum* GUANKE had a beneficial effect on alleviating constipation in STC mice. We speculate that the effect of *L. plantarum* GUANKE is probably mediated through different mechanisms, e.g., restored protein levels of constipation-related intestinal factors, rebalanced intestinal neurotransmitters and hormone release, and altered specific bacterial taxa, rather than an increase in the overall diversity of the gut microbiota. Further studies are necessary to elucidate the underlying mechanisms by which *L. plantarum* GUANKE modulates the gut microbiota in clinical trials and its therapeutic candidates for constipation treatment.

## Figures and Tables

**Figure 1 nutrients-15-03741-f001:**
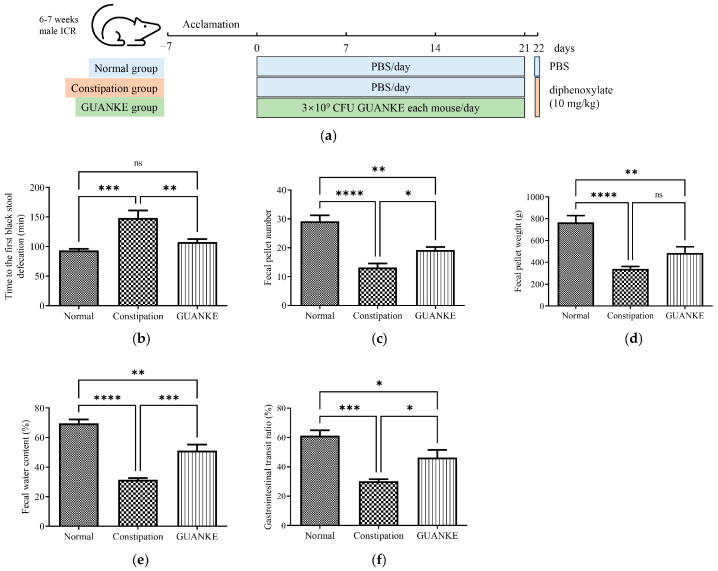
Effects of *L. plantarum* GUANKE on the parameters related to slow transit constipation in mice. (**a**) The experimental design workflow for ICR mice; (**b**) times to first black stool defecation; (**c**) fecal pellet number; (**d**) fecal pellet weight; (**e**) fecal water contents; (**f**) gastrointestinal transit. Data analysis was performed using one-way ANOVA, followed by LSD (least significant difference) for multiple comparisons test. **** *p* < 0.0001, *** *p* < 0.001, ** *p* < 0.01, * *p* < 0.05, ns = no significant difference *(p* > 0.05).

**Figure 2 nutrients-15-03741-f002:**
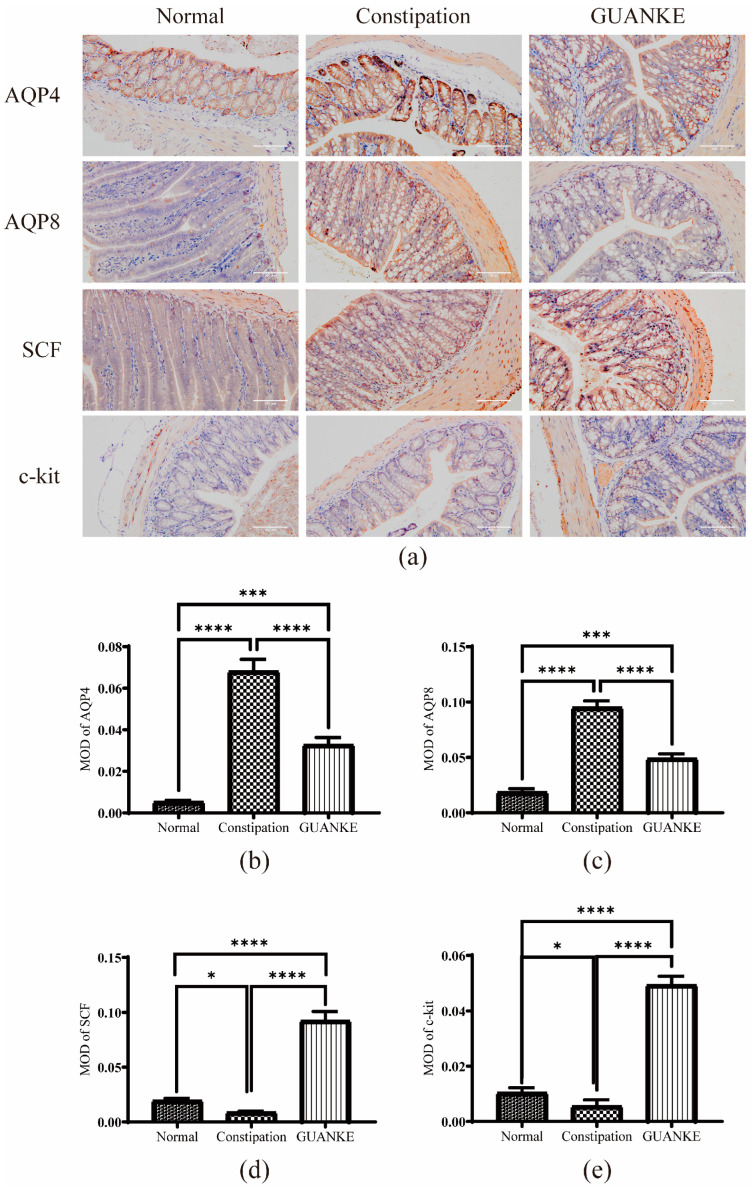
Effects of *L. plantarum* GUANKE on protein expression levels of intestinal factors associated with constipation in mice. (**a**) Immunohistochemical detection of aquaporins (AQP4 and AQP8) and interstitial Cajal cells (c-kit and SCF) in colons. Mean optical density (MOD) values were determined and analyzed for (**b**) AQP4, (**c**) AQP8, (**d**) SCF, and (**e**) c-kit. Statistical analysis was conducted using one-way ANOVA followed by multiple comparisons test using LSD (least significant difference) for each group. **** *p* < 0.0001, *** *p* < 0.001, * *p* < 0.05.

**Figure 3 nutrients-15-03741-f003:**
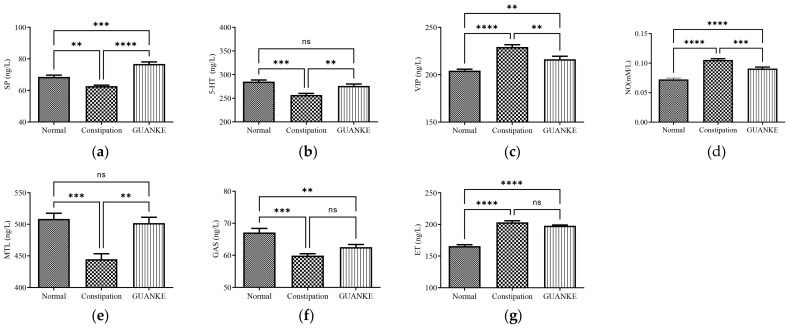
Effects of *L. plantarum* GUANKE on the content of gastrointestinal regulatory-related peptides in mice. (**a**) SP, substance P; (**b**) 5-HT, 5-hydroxytryptamine; (**c**) VIP, vasoactive intestinal peptide; (**d**) NO, nitric oxide; (**e**) MTL, motilin; (**f**) GAS, gastrin; (**g**) ET, endothelin. Data analysis was performed using one-way ANOVA, followed by multiple comparisons test for each group using LSD (least significant difference). **** *p* < 0.0001, *** *p* < 0.001, ** *p* < 0.01, ns = no significant difference (*p* > 0.05).

**Figure 4 nutrients-15-03741-f004:**
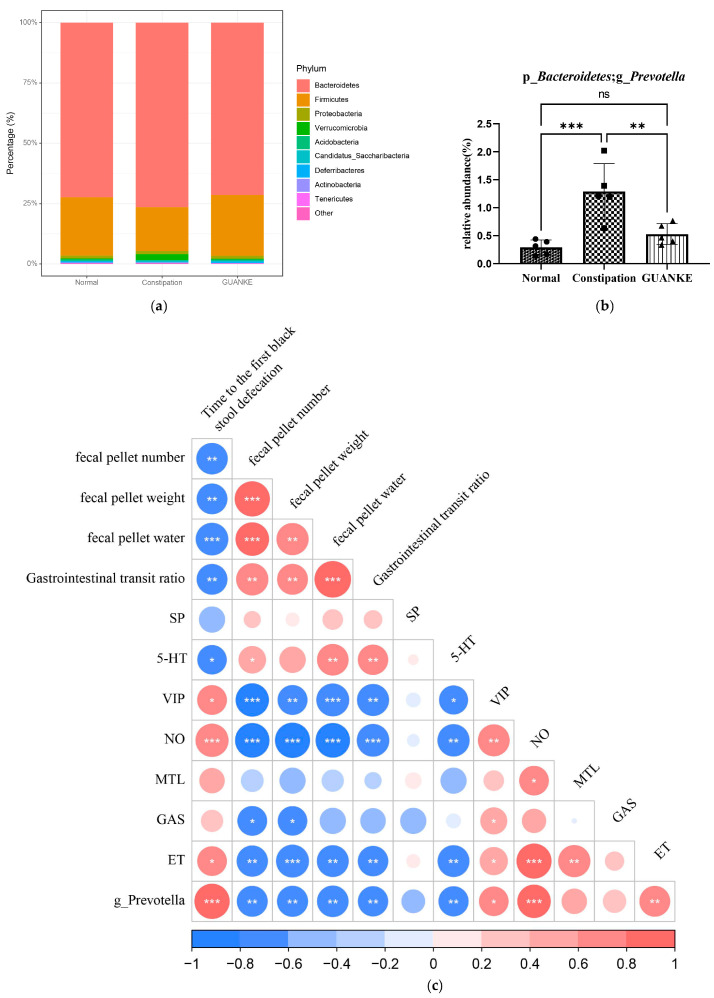
Effects and correlation analysis of *L. plantarum* GUANKE on gut microbiota in mice. (**a**) Taxonomic analysis illustrating the abundance of gut microbiota; (**b**) relative abundance of the differential bacterium *Prevotella* identified through LEfSe analysis; (**c**) correlation analysis of fecal parameters, intestinal neurotransmitters, hormones, and *Prevotella*. The nonparametric rank-sum test was used to compare the relative abundance of *Prevotella* in three groups. Correlation analysis of fecal parameters was performed by Pearson correlation analysis for fecal parameters, intestinal factors, and *Prevotella*. *** *p* < 0.001, ** *p* < 0.01, * *p* < 0.05, ns = no significant difference (*p* > 0.05).

**Figure 5 nutrients-15-03741-f005:**
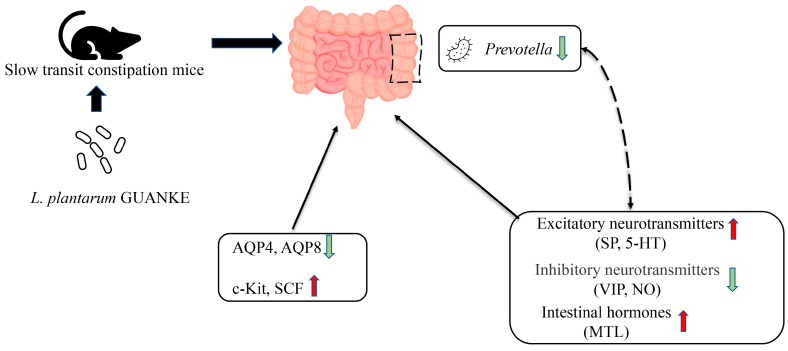
Potential mechanisms of *L. plantarum* GUANKE relieving slow transit constipation in mice. *L. plantarum* GUANKE administration restored protein levels of intestinal factors associated with constipation, increased the levels of 5-hydroxytryptamine, substance P, and motilin, and reduced levels of nitric oxide and vasoactive intestinal peptide. Moreover, GUANKE effectively reduced the relative abundance of *Prevotella*.

## Data Availability

The sequence information and clean data have been submitted to the NCBI SRA database under the following BioProject accession number: PRJNA977384.

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
