# Peer review of "Effects of Lactiplantibacillus plantarum GUANKE on Diphenoxylate-Induced Slow Transit Constipation and Gut Microbiota in Mice"

_nutrients, 2023, doi:10.3390/nu15173741_

Round 1

Reviewer 1 Report

This is very interesting study with extensive work using animal model and the results are of high interest to the reader in the area. The extesnise microbioal and biochemical analysis plus the relative transit ratio is a strong point. Only the methods (the design is hard to understand from the abstract also needed some time from me as a reader to figure it out. I recommend using a low chart diagram in the methodology section to clarify the desing. This would show timetimeline of the study groups and which one recievied which.

In addition to the improving the writing of the experimental desing, lines 171-173 remained by mistake and they should be deleted.

In line 82, should we say "a health volunteer" instead of "healthy people" unless the strain is a kind of isolated from mixed stools? If the strains was deposited in a culture collection, then that could be clarified too.

In line 68-72, a kind of repeated aims and worth considering either deleting the seconds sentence and changed the "symptom" into gut symptom in the first sentence or rephrasing that part. This is only because the word symptoms does not tell much and also the second sentence says "the results will confirm ...." which you don't know beforehand if the results would confirm and also relieving the STC was not mentioned in th eprevious sentence nor studied before.

In line 12, I think it is more accurate to say that "some probiotics can..." than "probiotics can" as not all strains can effectively relieve constipation.

In line 27, I suggest: are needed to elucidate if the mechanism obeserved here holds under other conditions or in human trials. 

Author Response

Q1: This is very interesting study with extensive work using animal model and the results are of high interest to the reader in the area. The extesnise microbioal and biochemical analysis plus the relative transit ratio is a strong point. Only the methods (the design is hard to understand from the abstract also needed some time from me as a reader to figure it out. I recommend using a low chart diagram in the methodology section to clarify the desing. This would show timeline of the study groups and which one recievied which.

Answer: Thank you for your insightful feedback and thoughtful suggestions on our study. We have taken your suggestion and added a flowchart diagram that outlines the timeline of the study groups and the treatments they received in Figure 1a. We are grateful that this visual aid will not only clarify the experimental design but also improve the overall reading experience of this manuscript.

Q2: In addition to the improving the writing of the experimental desing, lines 171-173 remained by mistake and they should be deleted.

Answer: Thank you for your valuable suggestion. We have revised the statement in lines 171-173, which was updated as: " Compared to the normal mice, the mice in constipation group exhibited longer defecation time expelling the first black stool, which indicated a weaker intestinal motility.

Q3: In line 82, should we say "a health volunteer" instead of "healthy people" unless the strain is a kind of isolated from mixed stools? If the strains was deposited in a culture collection, then that could be clarified too.

Answer: Thank you for this suggestion. We have corrected the "healthy people" into "a health volunteer" in line 82 in revised manuscript.

Q4: In line 68-72, a kind of repeated aims and worth considering either deleting the seconds sentence and changed the "symptom" into gut symptom in the first sentence or rephrasing that part. This is only because the word symptoms do not tell much and also the second sentence says, "the results will confirm ...." which you don't know beforehand if the results would confirm and also relieving the STC was not mentioned in the previous sentence nor studied before.

Answer: Thank you for your valuable suggestion. We have revised these sentences as follows: In this study, we systemically evaluated the effects of oral administration of L. plantarum GUANKE on constipation symptom, water channel protein expression and SCF/c-kit pathway in colon tissues by immunohistochemistry, quantification of neurotransmitter and hormone in serum, short-chain fatty acid abundance in cecum content, and fecal gut microbiota using diphenoxylate-induced constipation mouse model, which may provide evidence of utilization of probiotics in preventing constipation.

Q5: In line 12, I think it is more accurate to say that "some probiotics can..." than "probiotics can" as not all strains can effectively relieve constipation.

Answer: Thank you for the suggestion. We have corrected this in line 14 in revised manuscript.

Q6: In line 27, I suggest: are needed to elucidate if the mechanism obeserved here holds under other conditions or in human trials.

Answer: Thank you for your valuable suggestion. We have revised the sentence as follows: Further research is needed to confirm if the findings observed in this study will be consistent in other animal study or clinical trials in line 28-30 in revised manuscript.

Reviewer 2 Report

Original paper describing the effects of a probiotic strain on constipation in an experimental mouse model. The experiments were properly planned and the results obtained were described in a clear and logical way.

However, as a reviewer, allow me to make some comments and suggestions:

Abstract:

line 15 - please explain the abbreviation ICK.

In abstract there is no information which metabolites, neutransmitters and hormones were evaluated and where. There is no information on how the gut microbiome was assessed. There is also no reference to specific neutrotransmitter and hormone results. It should be added.

Introduction:

line 57 - the term "flora" should not be used nowadays rather microbiota or microbiome.

The Introduction section lacks information on the mechanism of constipation and the description of constipation-related intestinal factors (such a term was used in the description of the research results - subchapter 3.2, line 188)

The research assumptions described in the introduction are not sufficient. Line 70 only says that symptoms, serum metabolites (what? and neutrotransmitters, hormones?) were assessed. There is no information about the performed immunohistochemical tests of intestinal samples. It should be added.

Methods

The section "Immunohistochemistry examination" does not list which proteins/molecules were evaluated. Only the names of the antibodies are given, which should be referred to a specific protein/molecule.

The methodology for evaluating protein expression at the tissue level is not described, which should be included in this section, specifying a specific morphometric program (if any). It should be written what part of the colon was used for immunohistochemistry.

line 139 - Are all these proteins neurotransmitters? Is GAS, MTL neurotransmitters or local hormones? Change the subsection title (line 139)

Results

Figure 2 - what does the abbreviation MOD stand for? (description of the assessment method should be described in detail in the methods section)

Line 191 – what does it mean “chemical barrier”?

Discussion and Conclusions

Showing the potential mechanism of action of the probiotic strain in the form of Figure 5 de facto summarizes the discussion and therefore I believe that the discussion of this figure and the proposed mechanism of action should be included in the Discussion section (not in the conclusions).

Author Response

Q1: line 15 - please explain the abbreviation ICK.

Answer: We apologize for any confusion caused by abbreviation usage. In line 15 of the manuscript, "ICR mouse" refers to a strain of albino mice derived from the SWISS strain. This strain was initially selected by Dr. Hauschka and later distributed to various locations by the Institute of Cancer Research (ICR) in the USA to establish a fertile mouse line.

Q2: In abstract there is no information which metabolites, neutransmitters and hormones were evaluated and where. There is no information on how the gut microbiome was assessed. There is also no reference to specific neutrotransmitter and hormone results. It should be added.

Answer: Abstract: Thank you for your valuable suggestion. We have revised the abstract as you suggested:

Slow transit constipation (STC) is a prevalent gastrointestinal condition with slow transit, and some probiotics can effectively relieve constipation, but exact mechanisms have not been fully understood. In this study, we evaluate the impact of Lactiplantibacillus plantarum GUANKE (GUANKE) on diphenoxylate-induced slow transit constipation and speculate on the underlying mechanisms in ICR mouse model. Administration of L. plantarum GUANKE alleviated constipation indexes, including defecation time, faecal output and water content, and gastrointestinal transit ratio. In addition, GUANKE restored the proteins expression of constipation-related intestinal factors (aquaporins (AQPs) and interstitial Cajal cells (ICC)) in colon tissues measured by immunofluorescence staining, regulated the neurotransmitters and hormones, such as increased levels of 5-hydroxytryptamine, substance P and motilin, and decreased levels of vasoactive intestinal peptide and nitric oxide in serum by ELISA. 16S rRNA and correlation analysis of feces indicated that GUANKE administration effectively reduced constipation-induced Prevotella enrichment and suggested a potential contribution of Prevotella to diphenoxylate-induced STC in mice. GUANKE had no effect on SCFAs in cecum content. This study revealed that GUANKE may alleviate constipation in mice through regulating intestinal neurotransmitter and hormones release and altering specific bacterial taxa, rather than by affecting SCFAs and the diversity of microbiota in the gut. Further research is needed to confirm if the findings observed in this study will be consistent in other animal study or clinical trials.

Introduction:

Q3: line 57 - the term "flora" should not be used nowadays rather microbiota or microbiome.

Answer: Thank you for your suggestion. We have made the change from "intestinal flora" to "intestinal microbiota" in line 58 in revised version.

Q4: The Introduction section lacks information on the mechanism of constipation and the description of constipation-related intestinal factors (such a term was used in the description of the research results - subchapter 3.2, line 188)

Answer: Thank you for your suggestion. Thank you for your suggestion. With regard to the highlighted lack of information on the mechanism of constipation and the description of the intestinal factors associated with constipation, we have incorporated this content into the relevant section of the introduction. This enhancement results in a more comprehensive and well-rounded introduction to the topic, significantly improving the overall clarity and context of the manuscript.

Q5: The research assumptions described in the introduction are not sufficient. Line 70 only says that symptoms, serum metabolites (what? and neutrotransmitters, hormones?) were assessed. There is no information about the performed immunohistochemical tests of intestinal samples. It should be added.

Answer: Thank you for your valuable suggestion. We have taken your advice on board and made the necessary revisions. The revised version now reads as follows: In this study, we systemically evaluated the effects of oral administration of GUANKE on constipation symptom, water channel protein expression in colon tissues by immunohistochemistry,  quantification of neurotransmitter and hormone in serum, abundances of short-chain fatty acid in cecum content, and fecal gut microbiota using diphenoxylate-induced constipation mouse model, which may provide evidence of utilization of probiotics in preventing constipation.

Methods

Q6: The section "Immunohistochemistry examination" does not list which proteins/molecules were evaluated. Only the names of the antibodies are given, which should be referred to a specific protein/molecule.

Answer: Thank you for your insightful suggestion. We have taken your feedback into account and have updated the subtitle accordingly. We have incorporated the names of AQP4, AQP8, c-kit, and SCF in the "Immunohistochemistry Examination" section. Furthermore, we have included details about the specific target proteins/molecule that were detected in line 133-135 in revised manuscript.

Q7: The methodology for evaluating protein expression at the tissue level is not described, which should be included in this section, specifying a specific morphometric program (if any). It should be written what part of the colon was used for immunohistochemistry.

Answer: Thank you for your valuable suggestion. We have modified this section, and the revised version now reads as follow: After the colon tissues underwent final dehydration, washing, and mounting, their images were examined and recorded under a microscope and the mean op-tical density (MOD) values were calculated. The primary antibodies included anti-AQP4 antibody (Abcam, USA), anti-AQP8 antibody (Absin, China), anti-c-Kit antibody (CST, USA), and anti-SCF antibody (Absin, China), and the tissue sections were captured using a Computer Image Processing System equipped with a CMOS camera (OLYMPUS, Japan). The images were then analyzed using Image-Pro Plus software (Media Cybernetic, US). The positive immunostaining intensity and the corresponding tissue area were assessed through a combination of human-computer interaction. MOD was derived by calculating the ratio of integrated optical density to the area of tissue. These outcomes were evaluated by a trained individual who was unaware of the treatment conditions associated with the samples. The statistical analysis of the average optical density was performed using GraphPad Prism 5.

Q8: line 139 - Are all these proteins neurotransmitters? Is GAS, MTL neurotransmitters or local hormones? Change the subsection title (line 139)

Answer: Thank you for your suggestion. We have made the change from "Quantification measurement of neurotransmitters" to "Quantification measurement of serum neurotransmitters and hormones" in line 165 in revised version. The substances of interest encompass substance P (SP), 5-hydroxytryptamine (5-HT), vasoactive intestinal peptide (VIP), and nitric oxide (NO), all which function as neurotransmitters. Furthermore, motilin (MTL), gastrin (GAS), and endothelin (ET) operate as local hormones.

Results

Q9: Figure 2 - what does the abbreviation MOD stand for? (Description of the assessment method should be described in detail in the methods section)

Answer: Thank you for your valuable suggestion. In Figure 2, "MOD" now stands for Mean Optical Density, as indicated in the revised figure legend. Additionally, we've included an explanation of the acronym in the Methods section to provide further clarity in line 143 in revised manuscript.

Q10: Line 191 – what does it mean “chemical barrier”?

Answer: Thank you for your question. The term "chemical barrier" pertains to the protective mechanism existing within the gastrointestinal tract. This mechanism involves a diverse range of secretions that are crucial for safeguarding the intestines against harmful agents and ensuring the efficient transit of food and waste. Disorders related to the chemical barrier, resulting from neurological imbalances, encompass a range of anomalies. These include aberrant water channel protein expression, perturbed neurotransmitter secretion within the gastrointestinal tract, irregular hormone expression, diminished numbers of Cajal interstitial cells, and reduced levels of short-chain fatty acids. Such disruptions collectively hinder the smooth movement of stool through the intestine, as well as the lubrication necessary for the process. This, in turn, contributes to the occurrence of constipation.

To address these aspects, we have thoughtfully incorporated a section in the introduction of our work. This addition aids in providing readers with a comprehensive understanding of the connections between chemical barrier disturbances and the development of constipation. Your inquiry has played a pivotal role in enhancing the clarity and depth of our manuscript.

Discussion and Conclusions

Q11: Showing the potential mechanism of action of the probiotic strain in the form of Figure 5 de facto summarizes the discussion and therefore I believe that the discussion of this figure and the proposed mechanism of action should be included in the Discussion section (not in the conclusions).

Answer: Thank you for your thoughtful observation. We appreciate your keen insight into the structure of our manuscript. You're right that the potential mechanism of action of the probiotic strain, as presented in Figure 5, essentially encapsulates the essence of our discussion. We acknowledge your recommendation to ensure that the discussion of this figure and the underlying mechanism of action should rightfully be incorporated into the Discussion section, rather than being placed in the Conclusions section.

Reviewer 3 Report

The research was well planem and performed. Te obtained results are interesting and confirm the perticipation of intestinal bacterie in the pathogenesis  of functional disorders of the GIT.  They have  cognitive value and practical implications, but require confirmation in further studies, including clinical trial. The work can be published in presented form. However, I suggest including justification for using of Lactiplantibacillus plantarum  GUANKE rather than other strains of probiotic  bacteria.

Author Response

Q1: The research was well planem and performed. The obtained results are interesting and confirm the participation of intestinal bacteria in the pathogenesis of functional disorders of the GIT.  They have cognitive value and practical implications, but require confirmation in further studies, including clinical trial. The work can be published in presented form. However, I suggest including justification for using of Lactiplantibacillus plantarum GUANKE rather than other strains of probiotic bacteria.

Answer: In our preliminary research, we identified GUANKE as a strain of Lactiplantibacillus plantarum that exhibits robust probiotic attributes, including acid and bile salt tolerance, along with immunomodulatory effects [1]. Building upon these promising characteristics, we aimed to explore additional beneficial value of GUANKE to promote gut function. Therefore, for this study, we aimed to evaluate the effect of GUANKE as a probiotic candidate on promoting intestinal function and alleviating constipation.

  1. Xu, J.; Ren, Z.; Cao, K.; Li, X.; Yang, J.; Luo, X.; Zhu, L.; Wang, X.; Ding, L.; Liang, J.; et al. Boosting Vaccine-Elicited Respiratory Mucosal and Systemic COVID-19 Immunity in Mice With the Oral Lactobacillus Plantarum. Frontiers in Nutrition 2021, 8, 789242, doi:10.3389/fnut.2021.789242.